# The chromatin remodelers RSC and ISW1 display functional and chromatin-based promoter antagonism

Timothy J Parnell[1,2†], Alisha Schlichter[1,2†], Boris G Wilson[1,2], Bradley R Cairns[1,2*]

[1]Department of Oncological Sciences, Howard Hughes Medical Institute, University of Utah School of Medicine, Salt Lake City, United States; [2]Huntsman Cancer Institute, University of Utah School of Medicine, Salt Lake City, United States

**Abstract** ISWI family chromatin remodelers typically organize nucleosome arrays, while SWI/SNF family remodelers (RSC) typically disorganize and eject nucleosomes, implying an antagonism that is largely unexplored in vivo. Here, we describe two independent genetic screens for *rsc* suppressors that yielded mutations in the promoter-focused ISW1a complex or mutations in the 'basic patch' of histone H4 (an epitope that regulates ISWI activity), strongly supporting RSC-ISW1a antagonism in vivo. RSC and ISW1a largely co-localize, and genomic nucleosome studies using *rsc isw1* mutant combinations revealed opposing functions: promoters classified with a nucleosome-deficient region (NDR) gain nucleosome occupancy in *rsc* mutants, but this gain is attenuated in *rsc isw1* double mutants. Furthermore, promoters lacking NDRs have the highest occupancy of both remodelers, consistent with regulation by nucleosome occupancy, and decreased transcription in *rsc* mutants. Taken together, we provide the first genetic and genomic evidence for RSC-ISW1a antagonism and reveal different mechanisms at two different promoter architectures.

*For correspondence: brad.cairns@hci.utah.edu

†These authors contributed equally to this work

Competing interests: The authors declare that no competing interests exist.

## Introduction

Genomic DNA is packaged into chromatin, a dynamic material that exhibits numerous changes in post-translational modifications, composition, and protein interactions. One aspect of chromatin modulation involves the assembly or disassembly of chromatin through active remodeling, which can confer either occlusion or access to the DNA—a process that is associated with virtually all DNA-mediated transactions, including transcription, replication, and repair. Each remodeling action, either assembly or disassembly, is mediated (in part) by specialized ATP-dependent chromatin-remodeling complexes (*Vignali et al., 2000*; *Clapier and Cairns, 2009*; *Narlikar et al., 2013*; *Bartholomew, 2014*).

Certain chromatin remodelers align with these two general categories: those that restrict DNA access by chromatin assembly and organization and those that promote DNA access by chromatin disassembly and disorganization. This broad separation in function can be partially illustrated by studies of individual chromatin remodelers and their effects on gene expression (*Angus-Hill et al., 2001*; *Fazzio et al., 2001*; *Vary et al., 2003*); in general, remodelers associated with chromatin disassembly promote DNA access and gene expression, while remodelers associated with chromatin organization more often repress gene expression, though there are exceptions to this simplified view (e.g., increased accessibility can promote repressor access to chromatin).

The SWI/SNF family of chromatin remodelers provides a well-studied example of remodelers associated with nucleosome disorganization and/or disassembly. In yeast, the RSC chromatin-remodeling complex is an essential and abundant paralog of the canonical SWI/SNF remodeler (*Cairns et al., 1996*). The central subunit of RSC, Sth1, is a DNA-dependent ATPase that translocates

**eLife digest** The genome of an organism can contain hundreds to thousands of genes. However, these genes are not all active at the same time. Instead, mechanisms exist that control when genes are switched off or on. One such mechanism alters how tightly DNA is packaged into a structure called chromatin. To form chromatin, DNA is wrapped around histone proteins at different points along its length; these structures are known as nucleosomes. Once formed, chromatin can either adopt a tightly packed form that represses gene activity or a less compact form associated with gene activation.

The proteins that control how chromatin is packed are called 'chromatin remodelers'. These proteins work in complexes that either disassemble chromatin—for example, by repositioning nucleosomes or removing histones—or organize chromatin by replacing nucleosomes and making it more compact.

Studies in many organisms have identified two key families of chromatin remodelers. In yeast, the ISWI family of complexes assembles chromatin and a protein complex called RSC disassembles chromatin. Parnell, Schlichter et al. used a range of genetic techniques to investigate whether these two chromatin-remodeling complexes work against each other in a species of yeast called *Saccharomyces cerevisiae*. The results suggest that this is indeed the case.

Both the ISWI complex and the RSC complex bind to regions of DNA called promoters, which are found at the start of a gene and help to regulate its activity. Parnell, Schlichter et al. found that the RSC complex helps to activate genes by establishing or maintaining regions of nucleosome-poor chromatin at a promoter. The chromatin is relaxed in these regions, which allows the proteins that activate genes to access the DNA. This effect is partially counteracted by the ISWI complex, which repositions nucleosomes across the promoters to fill the gaps created by the RSC complex.

In comparison, Parnell, Schlichter et al. found that promoters that do not have regions of nucleosome-poor chromatin contain a larger number of both of the remodeling complexes and have a high turnover of histone proteins. This suggests that at these sites, the RSC proteins are needed to overcome the assembly of nucleosomes by the ISWI complex in order to activate the gene. Thus, these two chromatin remodelers, ISWI and RSC, compete at promoters to determine whether they contain or lack nucleosomes, which helps determine whether the gene is silent or active, respectively. Future work will look further at how the 'winner' is determined: how transcription factors or signaling systems within the cell bias the recruitment or activity of RSC or ISWI at particular promoters, to determine gene activity.

DNA, pumping DNA around the surface of a nucleosome, and effectively mobilizing the nucleosome with respect to the underlying sequence (*Saha et al., 2002*, *2005*). This property enables RSC to shift nucleosome positions, as well as completely eject nucleosomes (*Lorch et al., 1999*; *Boeger et al., 2004*; *Clapier and Cairns, 2009*; *Dechassa et al., 2010*). In vivo, RSC facilitates transcription by all three RNA polymerases, primarily by enabling promoter access (*Parnell et al., 2008*). RSC maintains proper promoter chromatin structure, as RSC mutants exhibit alterations in nucleosome occupancy and spacing at promoters (*Badis et al., 2008*; *Hartley and Madhani, 2009*; *Ganguli et al., 2014*). RSC activity appears regulated, in part, by the presence of histone modifications (*Kasten et al., 2004*; *Ferreira et al., 2007*). RSC contains seven bromodomains on four subunits, implying a key role of acetylation in regulation. Thus, gene activation often involves the recruitment and activation of remodelers such as RSC to act on specific modified nucleosomes and promote promoter accessibility. The converse of gene activation, silencing, is expected to be the reverse process, where nucleosomes are re-positioned and organized to occlude transcription factor access.

This reconfiguration of chromatin to a less active or repressive state is a function of other chromatin remodelers, including members of the ISWI family. In yeast, these include two highly conserved ATPase paralogs, *ISW1* and *ISW2*, related to the *Drosophila* 'Imitation SWitch' (ISWI) protein, which is the catalytic component of multiple chromatin-remodeling complexes with roles in nucleosome assembly and gene repression (*Tsukiyama et al., 1999*; *Vary et al., 2003*). Similar to the family of SWI/SNF remodelers, the ISWI family of remodelers uses DNA translocation to mobilize nucleosomes, though ISWI remodelers are typically restricted to movement/sliding only and not ejection

(*Whitehouse et al., 1999*; *Clapier and Cairns, 2009*). Importantly, ISWI generates regularly spaced nucleosome arrays by 'measuring' the length of DNA linker between nucleosomes, and this property is thought to enable gene repression by ordering nucleosomes into closely spaced regular arrays that can restrict access to DNA (*Grune et al., 2003*; *Whitehouse and Tsukiyama, 2006*; *Gangaraju and Bartholomew, 2007*; *Tirosh et al., 2010*; *Bartholomew, 2014*).

Studies of remodeler antagonism have been limited. ISW2 function was shown in one study to restrict the binding of the SWI/SNF chromatin remodeler at a target gene in yeast (*Tomar et al., 2009*). Another study showed antagonistic roles by two alternative assemblies of mammalian SWI/SNF complex (BRG and BRM), where BRM appeared to repress BRG activation functions (*Flowers et al., 2009*). A third noted attenuation of BRG activation by the CHD family remodeler Mi-2 (*Ramirez-Carrozzi et al., 2006*) at a set of target genes. Although notable, none of the prior studies provide a conceptual view of how two remodelers might antagonize one another at a large number of loci and how antagonism relates to nucleosome occupancy and positioning at co-occupied loci.

Here, we examine remodeler antagonism explicitly, providing the first evidence for an antagonistic relationship between ISWI and RSC. We demonstrate the suppression of growth rate phenotypes and the impact of these remodelers on both transcription and chromatin architecture at a genome scale. These studies uniquely reveal important activities of these two chromatin remodelers at particular promoter architectures—'open' and 'closed'—and the requirement for remodeler antagonism for proper regulation.

## Results

### A genome-wide screen for null suppressors of *rsc7Δ*

Rsc7 is a non-essential subunit of the RSC complex that is required for growth at elevated temperatures and for full growth under particular conditions (*Wilson et al., 2006*). We previously used a synthetic genetic array (SGA) screen (*Tong et al., 2001*) to identify genes that induced lethality in combination with *rsc7Δ* (*Wilson et al., 2006*). We again used the SGA array to screen for genes whose null mutation would allow for growth of *rsc7Δ* at an otherwise non-permissive temperature.

To accomplish this, a strain containing *rsc7Δ* was crossed to a haploid deletion library comprised 4700 strains, each bearing a deletion in a single nonessential gene (SGA array). Double mutants were isolated and screened for growth at a restrictive temperature. Four viable combinations were obtained, including combinations of *rsc7Δ* with *rpl20bΔ*, *lsm7Δ*, *bud31Δ*, or *isw1Δ* (*Figure 1A*, *Figure 2C* (33°C), and data not shown). The first three genes are associated with various ribonucleoprotein complexes, including ribosomes and snRNPs, which may represent interesting pathways that involve RSC function. However, the identification of *isw1Δ* was particularly intriguing, as it suggested a possible antagonistic relationship between ISWI complex(es) and RSC. Suppression was linked to *isw1Δ*, as the *rsc7Δ* temperature sensitivity returned with the introduction of a wild-type copy of *ISW1* on a plasmid (*Figure 1A*). The specificity of this observation is notable, as virtually all combinations of *rsc7Δ* with mutations in other chromatin-related genes typically resulted in lethality (*Wilson et al., 2006*); thus, *isw1Δ* was the sole suppressing mutation with a chromatin/transcription function isolated in our genome-wide format.

### A screen for suppressors of *RSC2* mutations yields suppressing mutations in histone H3 and H4

The suppression relationship between RSC and *ISW1* was further strengthened through a second independent genetic screen involving *RSC2*. The Rsc1 and Rsc2 proteins are two homologous mutually exclusive subunits of RSC that define two distinct RSC sub-complexes (*Cairns et al., 1999*). Loss of either separately confers distinct phenotypes, while loss of both is lethal, suggesting both unique nonessential and redundant essential functions within the complex (*Cairns et al., 1999*). Rsc1 and Rsc2 share the same domain structure, which consists of one nonessential and one essential bromodomain (*Cairns et al., 1999*), a BAH domain that binds histone H3 (*Tsankov et al., 2011*), and an AT hook (*Cairns et al., 1999*). We previously isolated mutant alleles in the BAH domain of Rsc2, including *rsc2-V457M* and *rsc2-D461G*, that confer temperature sensitivity in *rsc1Δ* strains (*Schlichter and Cairns, 2005*).

To identify whether histone mutations might suppress RSC mutations, we used a histone mutagenesis screen. We integrated the *rsc2-V457M* allele into an *rsc1Δ* strain bearing histone H3-H4 deletions (*hht1-hhf1Δ* and *hht2-hhf2Δ*) that was covered by a *URA3*-marked plasmid bearing

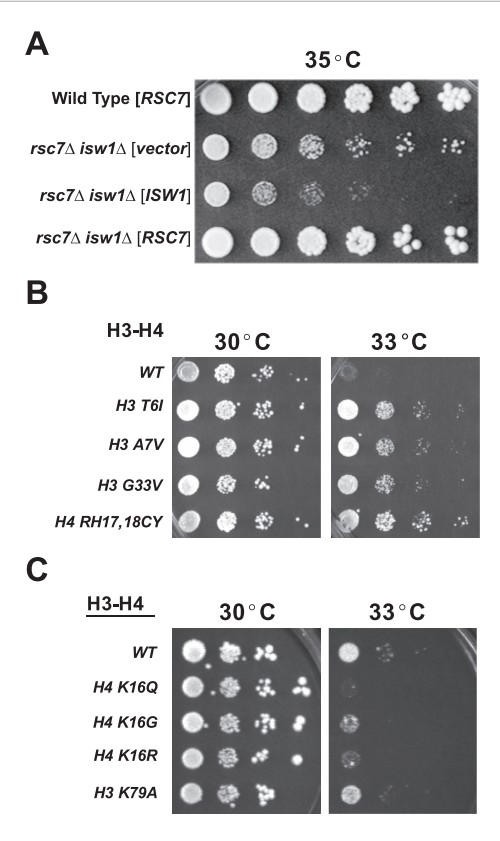

**Figure 1**. Suppressors of *rsc2* and *rsc7* alleles obtained by genetic screen. (**A**) Suppression of the *rsc7Δ* temperature sensitivity by the *isw1Δ* mutation. Wild-type (YBC62) and *rsc7Δ* (YBC2039) were transformed with plasmids containing *RSC7*, *ISW1*, or empty vector and spotted as fivefold serial dilutions to SC-URA media and grown at 35°. (**B**) Histone H3 and H4 suppressors of *rsc2-V457M*. YBC2140 (*rsc1Δ rsc2-V457M hht1hhf1 hht2-hhf2 [H3-H4.URA]*) was transformed with *TRP1*-marked plasmids bearing histone mutations, streaked to SC-TRP + 5FOA to force loss of the WT histone plasmid, and then spotted as 10-fold serial dilutions to SC-TRP at 30°C or 33°C. (**C**) 10-fold dilutions of YBC2140 transformed with H3 or H4 mutations and spotted to SC-TRP + 5FOA at 30°C or 33°C. *Figure 1—figure supplement 1* shows that Rsc2 alleles are not suppressed by *dot1Δ* or *sir3Δ*.

The following figure supplement is available for figure 1:

**Figure supplement 1**. *Rsc2* mutations are not suppressed by *dot1Δ* or *sir3Δ*.

wild-type histones, *HHT2-HHF2*. We then introduced *TRP1*-marked plasmids containing hydroxylamine mutagenized *HHT2-HHF2* genes and screened for suppression of the temperature sensitivity phenotype upon loss of the wild-type histone plasmid (using 5-FOA negative selection). From 20,000 transformants screened, we isolated seventeen suppressors that were verified by isolating and retransforming the plasmid containing the histone mutation. Of these, most contained single mutations: eight had either H3 A7V or H3 A7T mutations, seven had an H3 T6I mutation, and one bore an H3 G33V mutation. However, one mutant bore an H4 RH17,18CY double mutation (*Figure 1B*). All of these histone mutations were also tested for suppression of other temperature-sensitive RSC alleles, including *rsc2-D461G*, *rsc2-Y337H*, and *rsc4-2*, and each was suppressed (data not shown), suggesting that these mutations generally suppress RSC defects and not a specific defect of *rsc2-V457M*. We focused on the H4 RH17,18CY mutant for subsequent studies as it caused the most robust suppression (*Figure 1B*).

The H4 RH17,18CY mutations are adjacent to H4 K16, a residue whose acetylation serves as a mark for active chromatin (*Millar et al., 2004*). As RSC contains several bromodomains and may be regulated by histone acetylation (reviewed in *Josling et al., 2012*), we considered whether loss of H4 K16 acetylation may underlie the suppression. However, no H4 K16 acetylation was detected by Western blot in the H4 RH17,18CY mutant, but this could either be the result of loss of acetylation or failure of the antibody to recognize the mutated epitope. We therefore combined *rsc2-V457M* with H4 K16Q, H4 K16R, and H4 K16G mutants to determine if loss of K16 acetylation was responsible for the suppression. However, combining these mutants resulted in a slight synthetic sickness instead of suppression (*Figure 1C*), ruling out this simple model.

Notably, the H4 RH17,18CY mutations define the center of a region of the H4 tail referred to as the 'basic patch', an epitope of known importance for the binding and activity of several chromatin-modifying factors including Isw1, Sir3, and Dot1 (*Clapier et al., 2002*; *Fazzio et al., 2005*; *Fingerman et al., 2007*; *Altaf et al., 2007*; *Wang et al., 2013*). To test if the suppression by the basic patch mutation was due to an inability of Dot1 to bind or methylate H3K79, we combined *rsc2-V457M* with either an H3 K79A mutation or *dot1* null mutant. However, we observed no effect with the H3 K79A mutation (*Figure 1C*), and the combination with *dot1Δ* resulted in synthetic sickness (*Figure 1—figure supplement 1*). Additionally, combination with an *sir3Δ* failed to suppress (*Figure 1—figure supplement 1*). Taken together, our results point strongly to *ISW1* as the most likely candidate for RSC mutant suppression, tested further below.

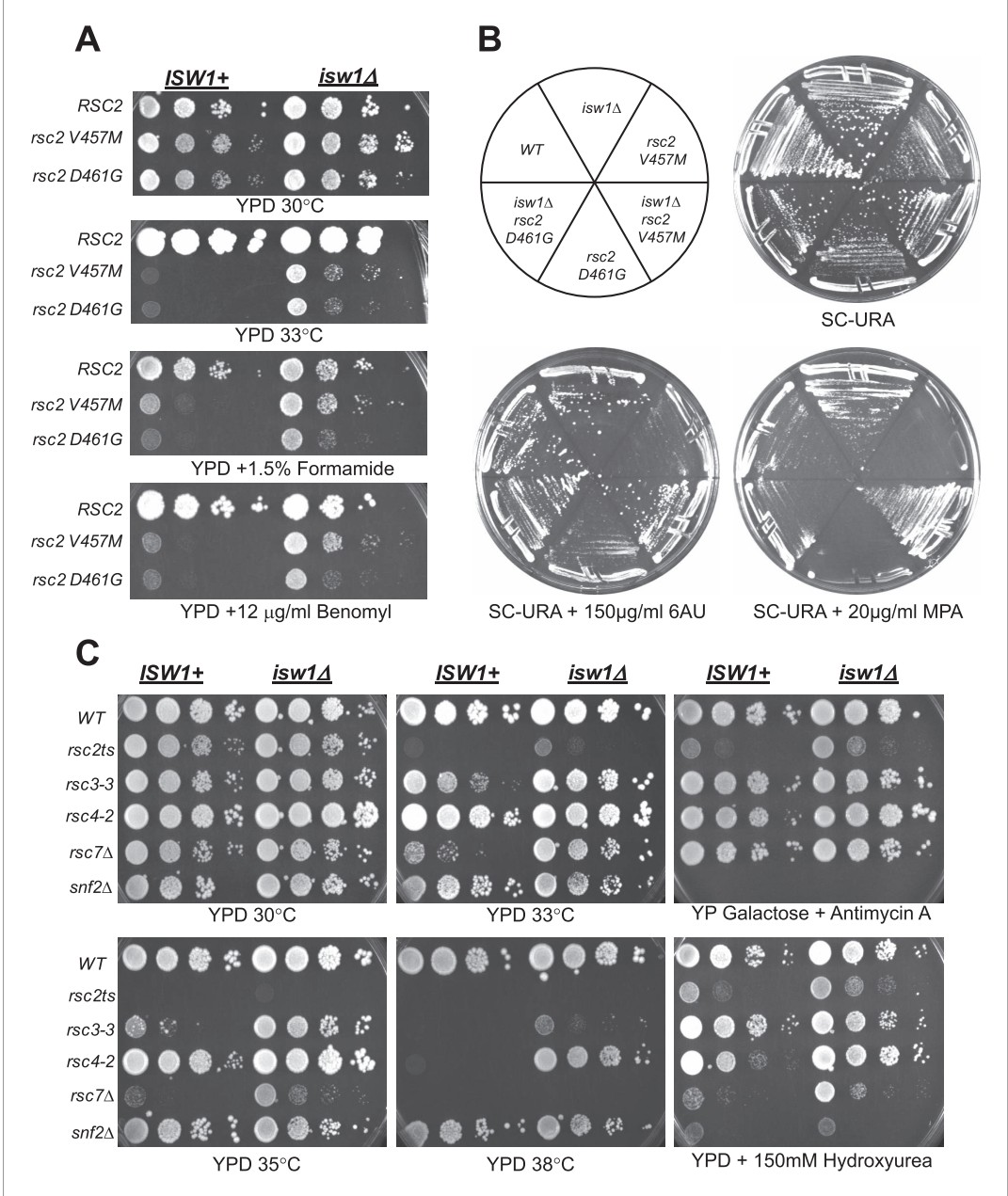

**Figure 2**. A null mutation of *isw1* suppresses RSC mutations. (**A**) *rsc2* Ts⁻ alleles are suppressed by *isw1Δ*. An *ISW1⁺* strain (YBC1231; *rsc1Δ rsc2Δ* [*RSC1.URA3*]) and an *isw1Δ* strain (YBC1479; *rsc1Δ rsc2Δ isw1Δ* [*RSC1.URA*]) were transformed with *TRP1*-marked RSC2 (p604), *rsc2-V457M* (p776), or *rsc2-D461G* (p777), streaked to SC-TRP + 5FOA to force loss of the RSC1 plasmid, and then spotted as 10-fold dilutions to YPD at 30°C, 33°C, and YPD containing 1.5% formamide (Form) or 12 μg/ml benomyl. (**B**) *isw1Δ* suppresses 6-azauracil (6AU) and MPA phenotypes of *rsc2* mutations. YBC1231 (*ISW1⁺*) and YBC1479 (*isw1Δ*) were transformed with *TRP1*-marked *RSC2* (p776), *rsc2-V457M* (p776), or *rsc2-*(YBC777), streaked to SC-TRP + 5FOA to force loss of the RSC1 plasmid, and then transformed with *URA3*-marked vector. Strains were then streaked to SC-URA medium containing 20 μg/ml MPA or 150 μg/ml 6AU. (**C**) *isw1Δ* suppresses additional RSC mutations but does not suppress *snf2*. WT (YBC62), *isw1Δ* (YBC1416), *rsc2-V457M* (YBC1111), *rsc2-V457M isw1Δ* (YBC2810), *rsc33* (YBC906), *rsc3-3 isw1Δ* (YBC1485 p817), *rsc4-2* (YBC1278), *rsc4-2 isw1Δ* (YBC2867), *rsc7Δ* (YBC1333), *rsc7Δ isw1Δ* (YBC2233), *snf2Δ* (YBC26), and *snf2Δ isw1Δ* (YBC2812) were spotted as 10-fold serial dilutions to YPD 30°C, 33°C, 35°C, 38°C, YPD containing 150 mM

*Figure 2. continued on next page*

*Figure 2. Continued*

Hydroxyurea (HU), and YPGal + Antimycin A (AA). *Figure 2—figure supplement 1* shows suppression of *rsc2* alleles by catalytic *isw1* and *isw2Δ* mutants.
The following figure supplement is available for figure 2:

**Figure supplement 1**. *rsc2* mutations are suppressed by an *ISW1* ATPase mutation and an *ISW2* null mutation.

## Loss of *ISW1* suppresses RSC mutations

The results of these two genetic screens strongly suggested a functional antagonism between RSC and Isw1. To directly test *isw1* mutant suppression of *rsc2* alleles, we combined *rsc2-V457M* or *rsc2-D461G* with *isw1Δ* and observed partial suppression of temperature sensitivity and a set of phenotypes associated with the drugs benomyl and formamide (*Figure 2A*) as well as 6-azauracil (6AU) and mycophenolic acid (MPA) (*Figure 2B*). Growth suppression of the double mutant was lost when *ISW1* was restored through plasmid transformation. Suppression requires a loss of ISW1 catalytic function, as *rsc2* suppression is observed in a strain bearing a mutation in the catalytic site (K227A) of ISW1 (*Figure 2—figure supplement 1A*). Furthermore, when we combined *rsc2-V457M* with both *isw1Δ* and H4 RH17,18CY, no enhanced suppression was observed (data not shown), suggesting that they act through the same pathway. We also directly tested whether the *ISWI* paralog, *ISW2*, might also suppress *rsc2* alleles. Combining the *isw2Δ* mutation with *rsc2-V457M* or *rsc2-D461G* did not confer suppression of the temperature growth defect, although some partial suppression of other phenotypes was observed (*Figure 2—figure supplement 1B*). We also did not see additional suppression when *isw1Δ* and *isw2Δ* were combined (data not shown). We therefore conclude that the *rsc2* mutation suppression is due primarily to the loss of Isw1 activity, with minimal contributions from of loss Isw2 activity.

We next asked whether *isw1Δ* suppression was specific to *rsc2* and *rsc7* mutations or could extend more generally to RSC mutations. To test this, we combined *isw1Δ* with two additional RSC mutations in separate subunits, *rsc3-3* and *rsc4-2*. The *isw1Δ* allele suppressed both RSC mutations tested (*Figure 2C*), with suppression of *rsc4-2* particularly robust and greater than *rsc2* mutants (*Figure 2C*, YPD 38°C panel). We also tested whether *isw1Δ* could suppress phenotypes associated with loss of SWI/SNF function. Combining *isw1Δ* with *snf2Δ* did not allow growth on galactose or raffinose carbon sources or growth on media containing hydroxyurea (*Figure 2C*), demonstrating specificity for RSC. Together, these results are consistent with a specific antagonistic relationship between RSC and *ISW1*.

## Suppression of *rsc2* by *isw1Δ* is specific to the ISW1a complex

Isw1 is the ATPase for two distinct remodeling complexes, ISW1a and ISW1b (*Vary et al., 2003*). The ISW1a form contains Ioc3, associates with particular gene promoters, and is implicated in repression by positioning nucleosomes into regularly spaced arrays (*Gangaraju and Bartholomew, 2007*; *Yamada et al., 2011*). ISW1b contains Ioc2 and Ioc4, associates with coding regions, plays a greater role in transcription elongation and termination, and does not regularly space nucleosomes (*Morillon et al., 2003*; *Vary et al., 2003*; *Gangaraju and Bartholomew, 2007*). To determine which form of the ISW1 complex is responsible for the suppression of *rsc2*, we combined *rsc2-V457M* with *ioc2Δ*, *ioc3Δ*, or *ioc4Δ*. Surprisingly, synthetic lethality, and not suppression, was observed when *rsc2-V457M* was combined with either *ioc2Δ* or *ioc4Δ* (*Figure 3A*). In contrast, combining *rsc2-V457M* with *ioc3Δ* resulted in partial suppression of *rsc2* phenotypes (*Figure 3B*). Notably, *ioc3Δ* potently suppressed *rsc4-2* (*Figure 3C*) and partially suppressed conditional *rsc1* mutations (*Figure 3—figure supplement 1*). Together, these results strongly implicate the loss of Isw1a complex function in *rsc* suppression.

## Loss of *isw1Δ* reduces the reliance of RSC on certain histone modifications

Our prior work revealed moderate *rsc2* suppression with increased H3K4me3 (by hyperactive *SET1* alleles), and conversely, synthetic lethality with *rsc2 set1Δ* combinations, suggesting that H3K4me3 either promotes or partially bypasses RSC activity (*Schlichter and Cairns, 2005*). However, as ISWI activity is affected by H3K4me and Set1 function (*Santos-Rosa et al., 2003*), an alternative hypothesis

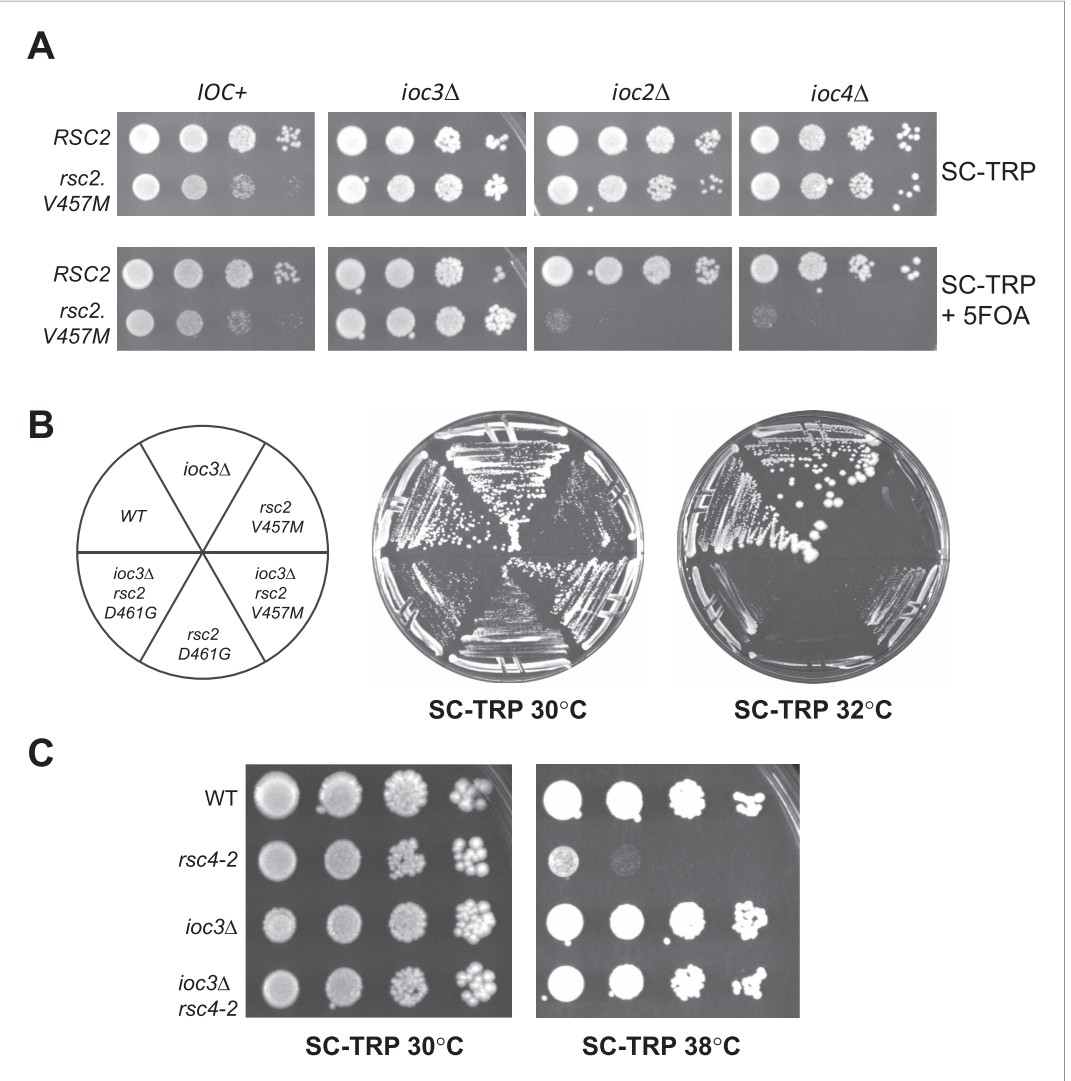

**Figure 3**. Suppression of *RSC* mutants is specific to Isw1a. (**A**) *rsc2-V457M* is lethal in combination with *ioc2Δ* and *ioc4Δ*, but not *ioc3Δ*. YBC803 (*rsc1Δ rsc2Δ* [*RSC1.URA3*]), YBC2730 (*rsc1Δ rsc2Δ ioc3Δ* [*RSC1.URA3*]), YBC2729 (*rsc1Δ rsc2Δ ioc2Δ* [*RSC1.URA3*]), and YBC2731 (*rsc1Δ rsc2Δ ioc4Δ* [*RSC1.URA3*]) were transformed with *TRP1*-marked *RSC2* (p604) or *rsc2-V457M* (p776), and spotted as 10-fold dilutions to SC-TRP 30°C or SC-TRP + 5FOA 30°C. (**B**) *rsc2* Ts mutations can be partially suppressed by *ioc3Δ*. YBC803 (*rsc1Δ rsc2Δ* [*RSC1.URA3*]) and YBC 2730 (*rsc1Δ rsc2Δ ioc3Δ* [*RSC1.URA3*]) were transformed with *TRP1*-marked *RSC2* (p604), *rsc2-V457M* (p776), or *rsc2-D461G* (p777), streaked to SC-TRP + 5FOA to force loss of the *RSC1* plasmid, and then streaked to SC-TRP at 30°C or 32°C. (**C**) *rsc4-2* is suppressed by *ioc3Δ*. Strain YBC627 (*rsc4* [*RSC4.URA3*]) and YBC3020 (*rsc4Δ ioc3Δ* [*RSC4.URA3*]) were transformed with *TRP1*-marked *RSC4* (p1060) or *rsc4-2* (p1083), streaked to SC-TRP + 5FOA to lose *RSC4.URA3*, and then spotted as 10-fold serial dilutions to SC-TRP 30°C or SC-TRP 38°C. **Figure 3—figure supplement 1** shows suppression of *rsc1* mutants by *ioc3Δ*. **Figure 3—figure supplement 2** shows the suppression of *rsc2* synthetic lethality with *set1Δ* and *gcn5Δ* by mutations in ISW1a.

The following figure supplements are available for figure 3:

**Figure supplement 1**. Mutations in RSC1 are suppressed by *ioc3Δ*.

**Figure supplement 2**. Synthetic lethality of *rsc2* mutations with *set1Δ* and *gcn5Δ* can be suppressed by *isw1* and *ioc3*.

is that H3K4me affects RSC indirectly through the alteration of Isw1 activity. To test this, we combined *rsc2-V457M* or *rsc2-D461G* with *set1Δ*, in the absence or presence of *ISW1* or *IOC3*. Interestingly, either *isw1Δ* or *ioc3Δ* can suppress the *rsc2 set1Δ* lethality (*Figure 3—figure supplement 2*). We also find that there is no additional suppression of *rsc2* temperature sensitivity by combining *SET1* hyperactive mutations and *isw1Δ* and that *isw1Δ* can still suppress *rsc2* phenotypes in an H3 K4A mutant (data not shown). These results suggest that suppression by loss of Isw1a is epistatic to the effects of Set1 loss and can overcome the reliance of RSC on H3K4 methylation.

As RSC activity is known to also be promoted by histone acetylation (e.g., H3K14ac; *Kasten et al., 2004*; *Carey et al., 2006*; *Ferreira et al., 2007*), we therefore tested whether loss of Isw1 would reduce the reliance of RSC on H3K14ac. *GCN5* is a histone acetyltransferase responsible for much of the H3K14ac in vivo (*Howe et al., 2001*; *Johnsson et al., 2009*), and loss of *GCN5* is lethal in combination with several RSC mutations, including *rsc2Δ* (*Cairns et al., 1999*; *Kasten et al., 2004*). We found *isw1Δ* suppressed the lethality of *rsc2Δ gcn5Δ* mutations (*Figure 3—figure supplement 2*). These results suggest that removing the chromatin remodeler that antagonizes RSC, notably ISW1a, reduces the need for RSC activation through acetylation.

## RSC and ISWI co-occupy many genomic locations

The genetic relationships identified above prompted us to investigate the spatial relationship between RSC and ISW1a. We therefore determined the occupancy of both of these chromatin remodelers by chromatin immunoprecipitation (ChIP), using the RSC subunit Rsc8 and the ISW1 subunit Ioc3, both tagged with C-terminal Myc epitope tags. We chose the Rsc8 subunit of RSC because it exists as a dimer in the RSC complex, minimizing the low ChIP efficiency observed with chromatin remodelers (*Whitehouse et al., 2007*; *Parnell et al., 2008*; *Yen et al., 2012*). We analyzed the immunoprecipitated DNA first by hybridization to high-resolution genome-wide microarrays (244K probes, ~50 bp resolution) and subsequently high-throughput sequencing.

RSC occupancy was scored across gene promoters (−800 to +800 bp), and promoters were then sorted into six clusters using a k-means algorithm to visualize those with and without enrichment (*Figure 4A*). Using the mean occupancy at the transcription start site (TSS, ±250 bp), 43% of promoters (2274 of 5337) had RSC enrichment corresponding to a false discovery rate (FDR) of less than 1%. We also found RSC was highly enriched at all non-coding RNA genes, including tRNA genes, as reported previously (*Ng et al., 2002*). Notably, we find RSC highly enriched at virtually all centromeres (*Figure 4B*), a localization not previously reported.

In comparison to RSC, the Ioc3 enrichment was less robust, perhaps reflecting a difference in chromatin association or difficulty in capturing complexes. We identified 137 or 230 Pol II promoters at an FDR of 1% or 5%, respectively. Strikingly, 224 of these latter promoters also pass the 1% threshold for RSC enrichment. Visual comparison of the enrichment pattern (log2 fold ChIP/Input) across all Pol II promoters reveals a high degree of overlap (*Figure 4A*), while a pairwise plot between the RSC and ISWI mean fold enrichment values at the TSS shows a positive correlation (r = 0.6; *Figure 4C*). This enrichment also extends beyond Pol II promoters, as we observed high ISW1a occupancy at both ncRNA and tRNA genes (*Figure 4B*). We did not observe significant enrichment of ISW1a at centromeres, although we note that ISW2 is enriched at centromeres (*Zentner and Henikoff, 2013*), which may provide any requisite ISWI function at these loci. These results support the notion that RSC and ISW1a share a spatial (though perhaps not temporal) occupancy at particular genes.

To extend these results, we repeated Rsc8, Ioc3, and Sth1 (the ATPase subunit of RSC) ChIP using micrococcal nuclease-digested chromatin analyzed by paired-end sequencing. We compared the log2 fold enrichment values obtained from both microarray and sequencing technologies (*Figure 4—figure supplement 1*). Despite the differences in resolution and sensitivity between these methods, we observed strong correlations between our microarray and sequencing results.

## Transcription-based suppression of *rsc2* by *isw1*

Since a complete loss of RSC function results in a cessation of all transcription from all three polymerases (*Parnell et al., 2008*), a weaker viable mutation (such as those in *rsc2*) may result simply in an attenuation of transcription of many or all genes, leading to a general phenotype such as temperature sensitivity. This transcription attenuation, as well as any suppression by ISWI, should be evident by expression analysis. To determine whether this suppression is global in nature or restricted

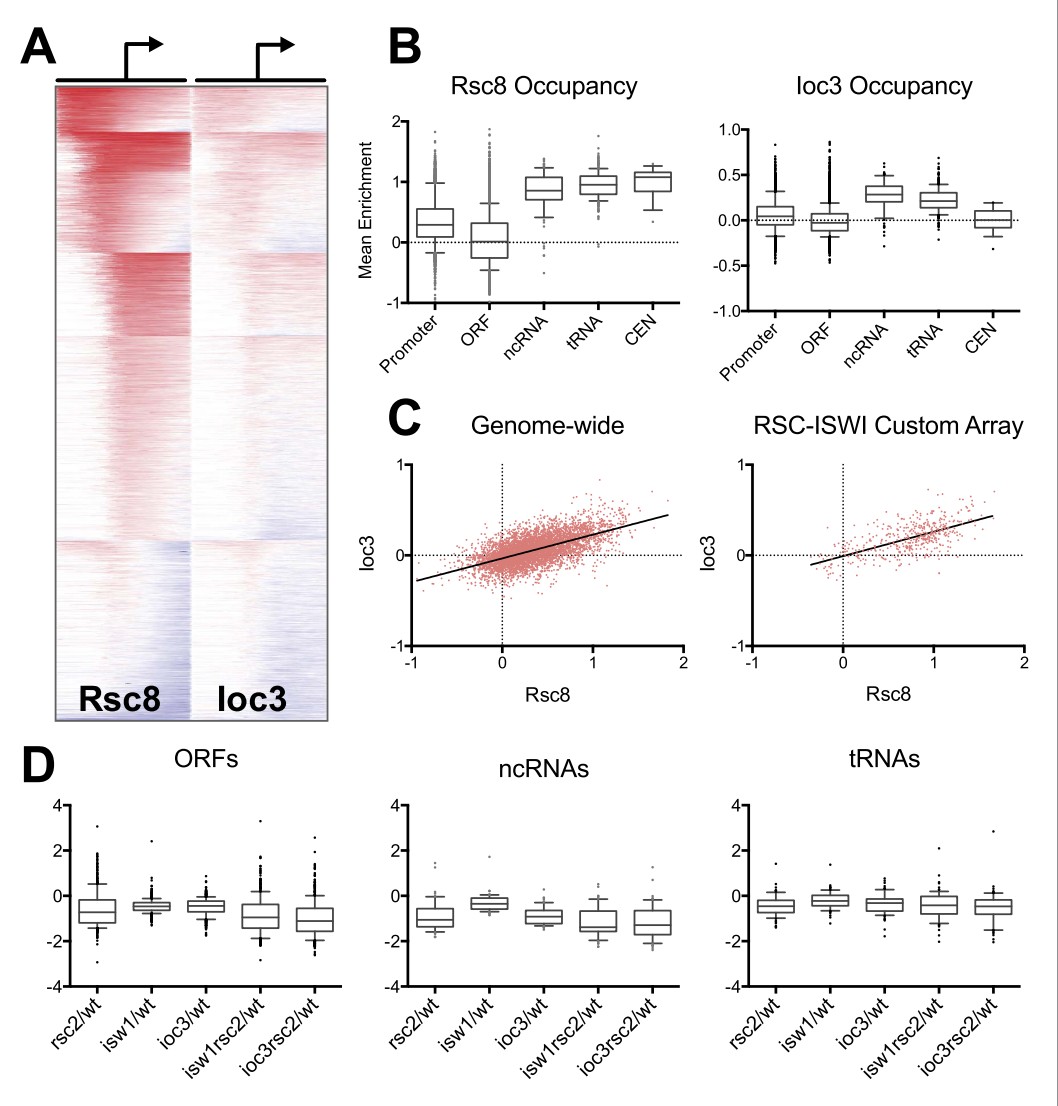

**Figure 4**. RSC and ISW1a co-occupy many locations, and their loss impacts gene expression in a complex manner. (**A**) Heat map of Rsc8 and Ioc3 protein occupancy as determined by ChIP at all TSS. Each row represents a gene, with occupancy scored in 50 bp windows, ±800 bp relative to the TSS (bent arrow). Windows overlapping neighboring genes are excluded. Occupancy above global mean is indicated in red, below in blue. Genes are clustered by a k-means algorithm into 6 groups. (**B**) The distributions of mean Rsc8 and Ioc3 occupancy values shown as box and whisker plots for different annotation features. (**C**) The correlation between Rsc8 and Ioc3 at promoters shown as a XY plot, either genome-wide or restricted to the 500 coding genes selected for the custom HybMap microarray. (**D**) The distribution of the mean mutant/wild-type gene expression ratios as determined by the HybMap microarray for three classes of gene types are presented as box and whisker plots. *Figure 4—figure supplement 1* compares the ChIP results obtained from microarray vs deep sequencing. *Figure 4—figure supplement 2* displays the genes that appear suppressed by *isw1Δ* or *ioc3Δ* as determined by HybMap.

The following figure supplements are available for figure 4:

**Figure supplement 1**. RSC and ISW1a occupancy correlate between microarray and sequence studies.

**Figure supplement 2**. Some genes show transcriptional suppression in *rsc2 isw1Δ* double mutants.

to a subset of genes, we performed a HybMap analysis on a sampling of genes in the genome. The HybMap technique measures both sense and anti-sense RNA levels across a genome (*Dutrow et al., 2008*), providing results that are comparable to RNA-Seq (*Ni et al., 2010*). The advantage of this

technique is the direct use of total RNA (enabling the detection of transcripts lacking polyA) without RNA labeling and/or amplification protocols to obtain absolute expression levels. Although the format restricted our array to 649 genes, it included a large fraction of genes occupied by RSC (84%), both RSC and ISW1 (9%), or unoccupied (16%), using an FDR threshold of 1%. We performed this analysis on *rsc2-V457M*, *isw1Δ*, *ioc3Δ*, *rsc2-V457M isw1Δ*, and *rsc2-V457M ioc3Δ* strains and compared them to wild type.

Consistent with the general requirement of RSC function for transcription (*Parnell et al., 2008*), the mean expression of both coding and non-coding genes (but not tRNAs) was reduced almost twofold following the loss of RSC (*Figure 4D*). Interestingly, individual *ioc3* or *isw1* mutants also lowered mean expression but with less magnitude. However, neither the *rsc2 ioc3* nor the *rsc2 isw1* double mutants generally suppressed the *rsc2* effect by restoring global gene expression. Furthermore, we saw little change among tRNA genes from any genotype and no measureable change in anti-sense transcription levels (data not shown). We also did not observe general aberrant transcription from promoters as reported previously in an *rsc3* mutant (*van Bakel et al., 2013*). These results suggest that the suppression of RSC phenotypes is not due to a global effect on gene expression but rather due to an effect at a subset of genes. To see if such genes could be identified from our sampling, we selected genes whose expression was at least partially restored by combining *rsc2* with *isw1* or *ioc3* mutations. This analysis revealed 20 genes (*Figure 4—figure supplement 2*), which included genes for ribosome function, snoRNA genes, and several essential genes. It is likely that the combined modest change in expression at these and other genes are responsible for the suppression relationship observed.

## ISW1 mutations suppress nucleosomal shifts in RSC mutants

Since RSC and ISW1 are both chromatin remodelers, the most important test for antagonism involves examining whether mutations in *ISW1* could suppress the effects of nucleosomal changes due to the loss of RSC function. Loss of RSC results in a gain of nucleosome occupancy at the nucleosome-depleted region (NDR) commonly found near the TSS of genes (*Badis et al., 2008*; *Parnell et al., 2008*; *Hartley and Madhani, 2009*; *Ganguli et al., 2014*). We therefore constructed strains that included the *sth1^td* degron allele in combination with an *isw1Δ* allele. Implementation of the *sth1^td* allele allows for precise inducible destruction of the catalytic subunit of RSC, thus abrogating all RSC catalytic function—which we subsequently term 'rscΔ' in figures and text. We chose to use both the RSC and *isw1* null alleles to maximize the nucleosomal effects due to the loss of catalytic activity in a manner that mutations in regulatory subunits may not. Mono-nucleosomal DNA was isolated from these yeast strains after inducing the degron allele for 2 hr and analyzed by both high-resolution microarray (rscΔ and *isw1Δ*) and paired-end sequencing (rscΔ only). As a reference, we also analyzed mono-nucleosomal DNA from control strains that cannot degrade Sth1 protein. To analyze the chromatin structure around the TSS, we generated nucleosome profiles around the TSS for every promoter by scoring the nucleosomal occupancy for rscΔ and RSC strains. Promoters were organized into clusters based on their rscΔ/RSC ratio profile using a k-means algorithm (*Figure 5A*). For each cluster, the mean nucleosome profile of both strains was then generated (*Figure 5B*) (We note that the clusters in *Figure 5* bear no relationship to the clustering analysis in *Figure 4*, which instead shows similarity in loci occupied by RSC or ISWI).

The aggregate nucleosome profiles of wild type (blue line, *Figure 5B*) confirmed published observations (*Yuan et al., 2005*; *Lee et al., 2007*; *Whitehouse et al., 2007*), showing a clear NDR flanked by positioned nucleosomes (termed −1 and +1) and phased positioned nucleosomes within the proximal coding region. The loss of RSC (*Figure 5A* and *Figure 5B*) resulted in two major categories: (1) clusters 1–4 all share strong changes in nucleosome positioning following the loss of RSC and (2) clusters 5 and 6 show a weak response to the loss of RSC. Closer examination revealed further differences in each category. For example, the cluster 1 rscΔ profile shows a dramatic gain in nucleosome occupancy over the NDR at the expense of the +1 nucleosome relative to the control RSC profile, consistent with prior work (see 'Discussion'). There is also a clear 'leftward' shift in nucleosome positions over the body of the gene, towards the NDR. This nucleosome shift is particularly prominent in clusters 3 and 4, while the NDR is filled into a lesser extent and the +1 nucleosome peak is not as depleted (compared to cluster 1). To confirm that these effects were not due to limitations in the sensitivity and resolution of microarray analyses, the nucleosome profiles from the microarray data were directly compared to those from the sequencing analysis (*Figure 5—figure supplement 1*). Nucleosome profiles of the same clusters show remarkable similarity between those derived from

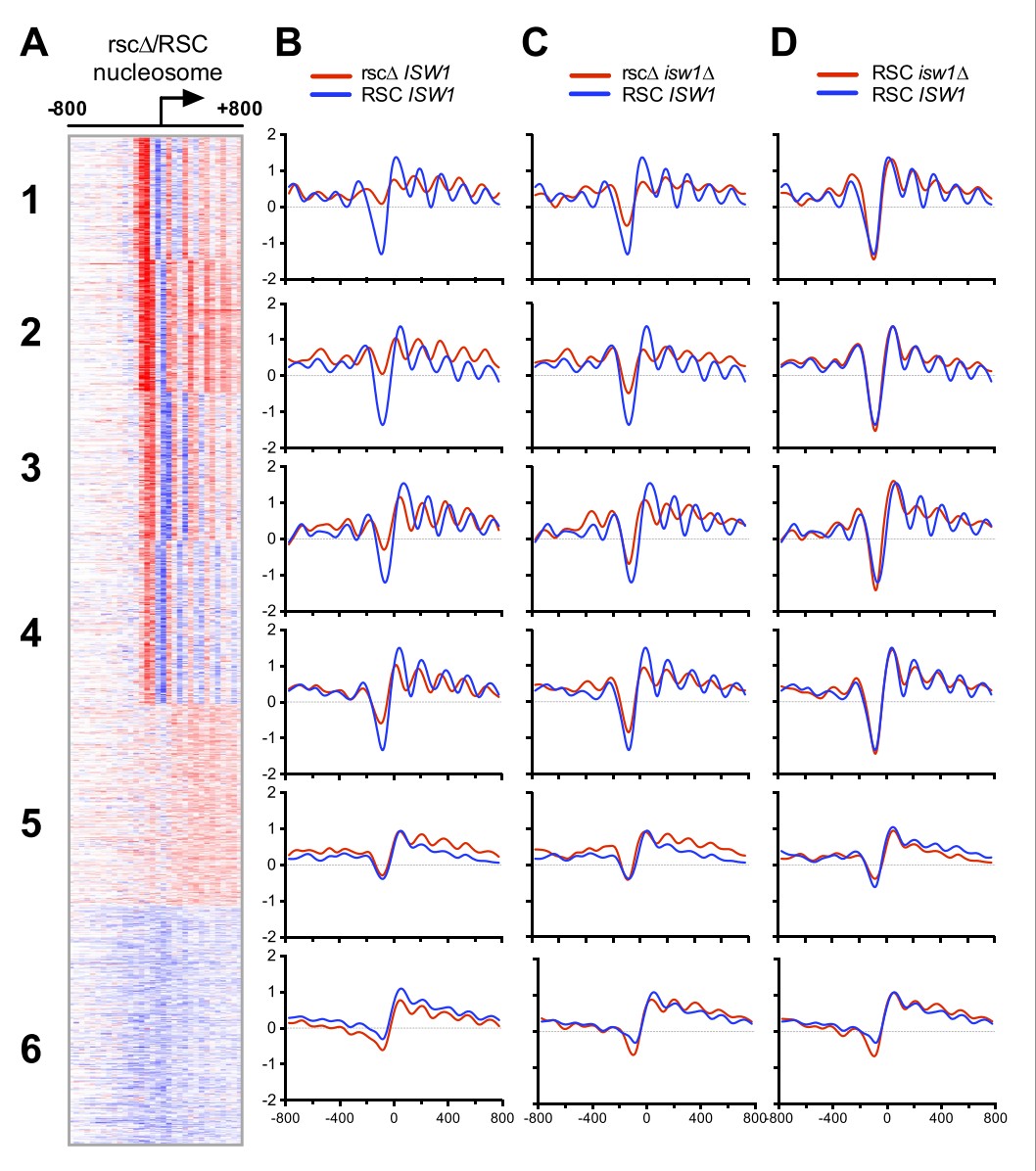

**Figure 5**. Loss of ISW1 partially suppresses nucleosomal changes exhibited by loss of RSC function. (**A**) The promoter profile of nucleosome occupancy ratios between *sth1^td* degron (rscΔ) and control (RSC) strains is presented as a heat map, where red represents a gain in nucleosome occupancy and blue represents a loss. Genes (rows) are organized into six groups by k-means clustering. Columns represent 50 bp windows, ±800 bp relative to the TSS. Windows overlapping neighboring genes are excluded. (**B**, **C**, **D**) The mean profiles of nucleosome occupancies for all genes within each cluster are shown. Profiles from mutant backgrounds are shown in red, and wild-type profiles are shown in blue. The y-axis represents log2 occupancy relative to genome average. *Figure 5—figure supplement 1* compares the nucleosome profiles obtained from microarray and deep sequencing, as well as the predicted nucleosome occupancy.

The following figure supplement is available for figure 5:

**Figure supplement 1**. Nucleosome profiles from microarray and sequencing show strong correlation with each other and predicted occupancy.

array and sequence, validating our approaches and conclusions. Taken together, the filling of the NDR and a strong 'leftward' shift of the +1 nucleosome toward the NDR are consistent features that follow loss of RSC function.

We next examined the impact due to the loss of *ISW1* (*Figure 5D*). Loss of *ISW1* results in modest nucleosomal changes, most notably within the promoter-proximal 5′ coding region, either as changes in density or phasing, and minimal impact at the NDR. While loss of *ISW1* alone has been shown to result in nucleosomal shifts towards the TSS (*Tirosh et al., 2010*; *Yen et al., 2012*; *van Bakel et al., 2013*), these shifts, discernable in cluster 3, are much smaller and more restricted than those generated by the loss of RSC (compare *Figure 5B,D*). Importantly, in the double mutant (*Figure 5C*), the nucleosomal profiles are more similar to wild type than rscΔ alone. Notably, the NDRs are not as filled and the shifts towards the TSS are not as severe. Taken together, these results provide considerable support for an antagonistic relationship between RSC and ISW1, especially regarding the positioning and phasing of nucleosomes over the promoter-proximal coding region of the gene.

## RSC loss impacts nucleosome structure at structured/open promoters more than unstructured/closed promoters

Above, we showed that nucleosome architecture at clusters 1–4 shows a strong response to RSC loss, whereas clusters 5 and 6 show apparently limited changes. Clusters 1–4 display a prototypical promoter nucleosomal architecture (−1, NDR, and +1 nucleosome). In contrast, clusters 5 and 6 lack this stereotypical organization; here, RSC and/or ISW1a may indeed impact nucleosome occupancy and/or positioning, but the effect may be obscured due to architectural heterogeneity. Notably, these two types of architectures have previously been designated as open (or structured) vs closed (or unstructured) and have been largely correlated with either constitutive or highly regulated gene types, respectively (*Tirosh and Barkai, 2008*; *Cairns, 2009*). We verified these classifications by plotting the mean nucleosome prediction (*Segal et al., 2006*) for each of these clusters (*Figure 5—figure supplement 1B*). While the predictive power for individual nucleosome positions was weak, the algorithm predicted the depth and breadth of NDRs fairly accurately. The strongly responsive clusters 1–4 had a well-defined NDR prediction, matching the observed profile, while the weakly responsive clusters 5 and 6 showed a broad shallow NDR. Since nucleosome phasing is, in part, determined by how well the −1 and +1 nucleosomes are positioned flanking the NDR, this result matches well with the general lack of consistent phasing across clusters 5 and 6 gene bodies. Interestingly, cluster 3 does not show as strong a predictive NDR as clusters 1, 2, and 4, which may partly explain why this cluster shows nucleosomal shifts in both isw1Δ and rscΔ and weak suppression in the double mutant. Taken together, structured/open promoters show the strongest response to RSC loss, whereas unstructured/closed promoters lack a strong response—though we note that the lack of a uniform structure may obscure the response (see 'Discussion').

Given the strong impact on chromatin structure at open/structured promoters vs the closed/unstructured promoters, we next examined how the loss of RSC might impact the transcription of these classes. Using our HybMap RNA expression data as a proxy for transcriptional impact, we scored genes from each category for expression. We note, however, that the differences between the HybMap and nucleosome experiments in several parameters, for example, RSC mutation vs depletion, time points, and representation among clusters (see *Figure 6—figure supplement 1A*), place limitations on these comparisons. Nevertheless, while all promoter classes showed reduced gene expression in *rsc2* mutants (consistent with *Figure 4D*), cluster 2 and especially clusters 5 and 6 (the two 'closed' promoter clusters) were most severely negatively impacted (*Figure 6A*). The inclusion of cluster 2 with clusters 5 and 6 is intriguing; however, it is also the only structured cluster to exhibit significant nucleosome occupancy gain over the body of the gene in rscΔ, which is likely related to the reduction in transcription. These impacts are not simply correlated with the level of gene expression, as the distribution of wild-type expression values between clusters is highly consistent (*Figure 6—figure supplement 1*). Similar to what we observed previously, the isw1Δ mutants showed little impact on the bulk expression of these genes (*Figure 6A*). We also examined the chromatin structure of the 17 Pol II-transcribed genes where ISW1a loss provided significant suppression of *rsc2*Δ (*Figure 4—figure supplement 2*). However, the moderate resolution provided by the microarray format limited the fine mapping of nucleosomes, preventing our ability to identify nucleosomes that might be directly responsible for suppression (data not shown). These 17 genes partitioned slightly more to closed (nine genes) than the more common open promoter structures (eight genes). Together, these results suggest that while the clearest effects on nucleosome positioning (restoration in isw1Δ) are seen with 'open' promoters, the largest effects on transcription are more closely associated with 'closed' promoters (*Figure 6A*).

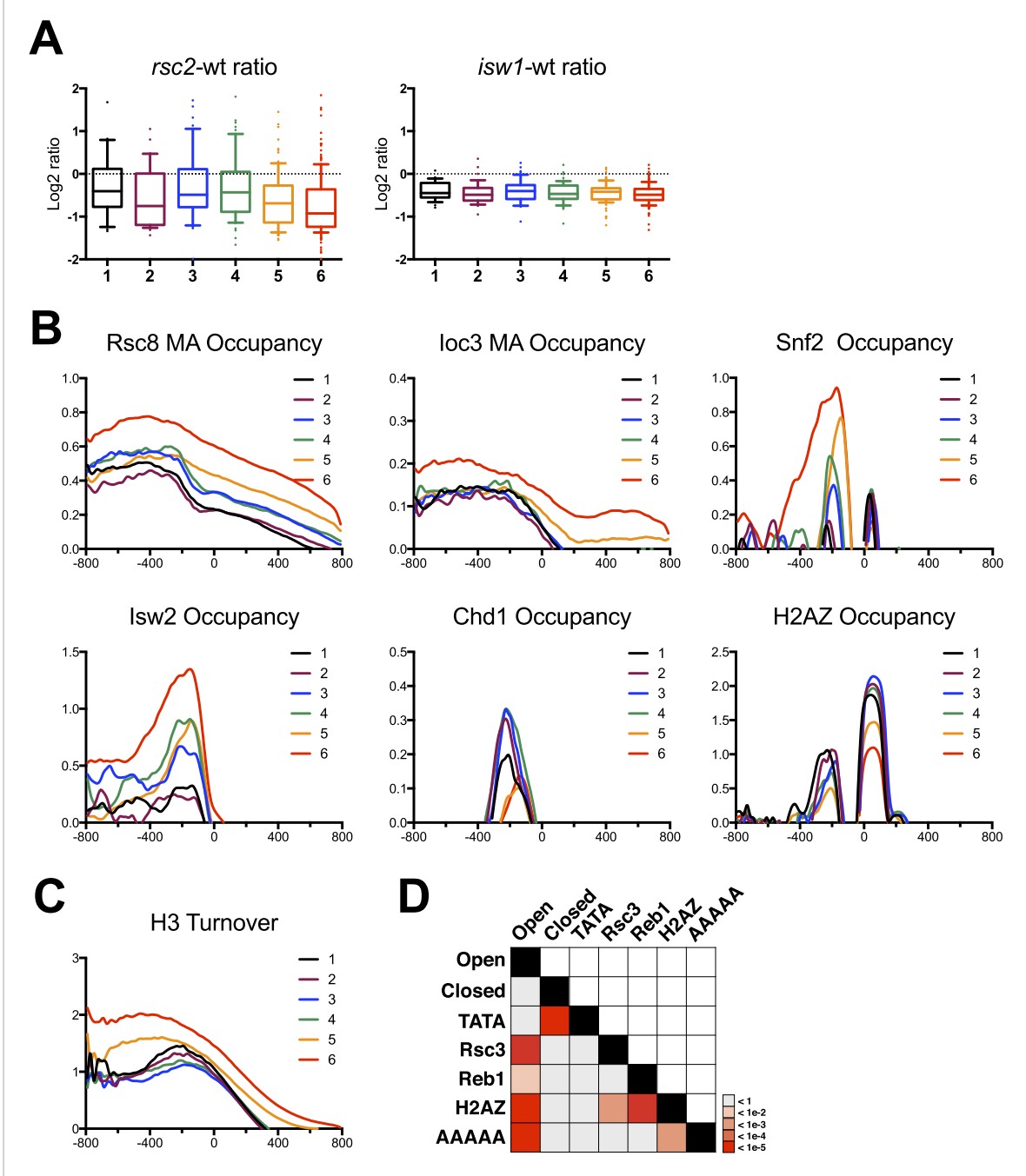

**Figure 6**. Gene clusters identified by their response to RSC loss reveal different promoter classes. (**A**) The relative expression in *rsc2* (left) or *isw1Δ* (right) mutants relative to wild type as measured by the HybMap assay are plotted as box and whisker distribution plots for each of the six gene clusters identified by their response to RSC loss. (**B**) The mean occupancy profile over each of the six gene clusters is presented for six different factors, including RSC, ISW1, SNF2, ISW2 (*Zentner and Henikoff, 2013*), CHD1 (*Zentner and Henikoff, 2013*), and H2AZ (*Albert et al., 2007*). (**C**) The mean profile for histone turnover over the six gene clusters is shown. Higher values represent higher turnover. (**D**) Heat map representing the p-value significance for the intersection between genes in different categories. Open promoters include genes in clusters 1–4. Closed promoters include genes in clusters 5 and 6. *Figure 6—figure supplement 1* shows the distribution of normal gene expression for each of the clusters. *Figure 6—figure supplement 2* shows the occupancy profile for RSC and ISW1a as determined by deep sequencing.

The following figure supplements are available for figure 6:

*Figure 6. continued on next page*

*Figure 6. Continued*

**Figure supplement 1**. Wild-type RNA expression levels are not significantly different between the six clusters.

**Figure supplement 2**. Occupancy of RSC and ISW1a as measured by sequencing.

## Unstructured/closed promoters have the highest RSC occupancy

We next addressed the relationship between RSC occupancy and promoter architecture (open vs closed promoters). Here, we plotted the mean occupancy profile for both Rsc8 and Ioc3 (*Figure 6B* and *Figure 6—figure supplement 2*) across the promoter for each of the six clusters identified in *Figure 5*. One might expect that genes with a strong response in regard to nucleosome positioning would have high RSC occupancy. Somewhat surprisingly, we found the opposite result. Clusters 5 and especially 6 had the highest mean occupancy of both RSC and ISW1a. We also examined other chromatin remodeler occupancies, including SWI/SNF (this study), Isw2 (*Zentner and Henikoff, 2013*), and Chd1 (*Zentner and Henikoff, 2013*). Notably, Isw2 and SWI/SNF occupancy displayed higher occupancy in cluster 6, while Chd1 was more equally distributed among the clusters. Interestingly, histone H2AZ demonstrated an inverse relationship, as clusters 5 and 6 bore the least H2AZ. Considering that these two gene clusters have the highest occupancy of chromatin remodelers, we next asked whether these genes also exhibited high histone turnover. We plotted the mean profile of measured histone turnover rate (*Rufiange et al., 2007*) over the six gene clusters and found that the degree of histone turnover correlated well with remodeler occupancy, with cluster 6 having the highest turnover, particularly around and upstream of the TSS (*Figure 6C*). Thus, unstructured/closed promoters have the highest remodeler occupancies and the highest turnover.

Together, these observations coalesce around the idea that these gene clusters identified in *Figure 5*, based solely on the impact of RSC remodeler loss, also broadly segregate genes into two distinct types of promoter architectures: open (structured) promoters and closed (unstructured). The gene clusters with the greatest measurable impact on chromatin organization due to RSC loss, groups 1–4, represent the open promoters, whereas groups 5 and 6 represent closed promoters, which collectively lack a distinctive organization and therefore a measurable impact. These promoter architectures matched well with the predictions of remodeler occupancy and histone turnover (*Tirosh and Barkai, 2008*; *Cairns, 2009*). These architectures are also predicted to correlate with specific DNA sequence characteristics and nucleosome composition. For example, open promoters typically contain nucleosome exclusion sequences clustered with binding sites for factors that may help exclude or reposition nucleosomes (*Segal et al., 2006*; *Badis et al., 2008*; *Hartley and Madhani, 2009*). To verify these, we scored promoters for the presence of TATA, Reb1, and Rsc3 binding sites, as well as the number of AAAAA sequences, which antagonize nucleosome formation (*Kaplan et al., 2009*; *Segal and Widom, 2009*). We then calculated the statistical enrichment of these sequence attributes for both structured (clusters 1–4) and unstructured (clusters 5–6) genes over background by random permutation analysis. We found that clusters 1–4 showed statistically significant enrichments for Reb1 and Rsc3 binding sites and AAAAA sequences, while closed promoters showed an enrichment of TATA binding sites, matching the predictions (*Figure 6D*). Importantly, it is unstructured/closed and TATA-rich promoters that have been shown previously to mostly rely on chromatin modifiers and remodelers for their activation (*Raser and O'Shea, 2004*; *Musladin et al., 2014*). As developed in the 'Discussion', we believe our results, combined with others, argue for two different modes of impact of RSC and other remodelers at the two promoter types: open and closed (*Figure 7*).

## Discussion

Chromatin remodelers represent a set of complexes with different functional roles; some remodelers are primarily involved in transcriptional activation, while others are more dedicated to chromatin assembly and/or transcriptional repression. Here, we describe an antagonistic relationship between two such chromatin remodelers, RSC and ISW1, through a combination of genetics, gene expression, and genome-wide nucleosome positioning studies. At genes, RSC is primarily utilized for gene activation, providing this function, at least in part, by establishing or maintaining the NDR structure at promoters. We find that this function is partly counteracted by Isw1 activity, which re-positions nucleosomes to 'fill in' the NDR and positions nucleosomes over *cis* regulatory sequences. While there

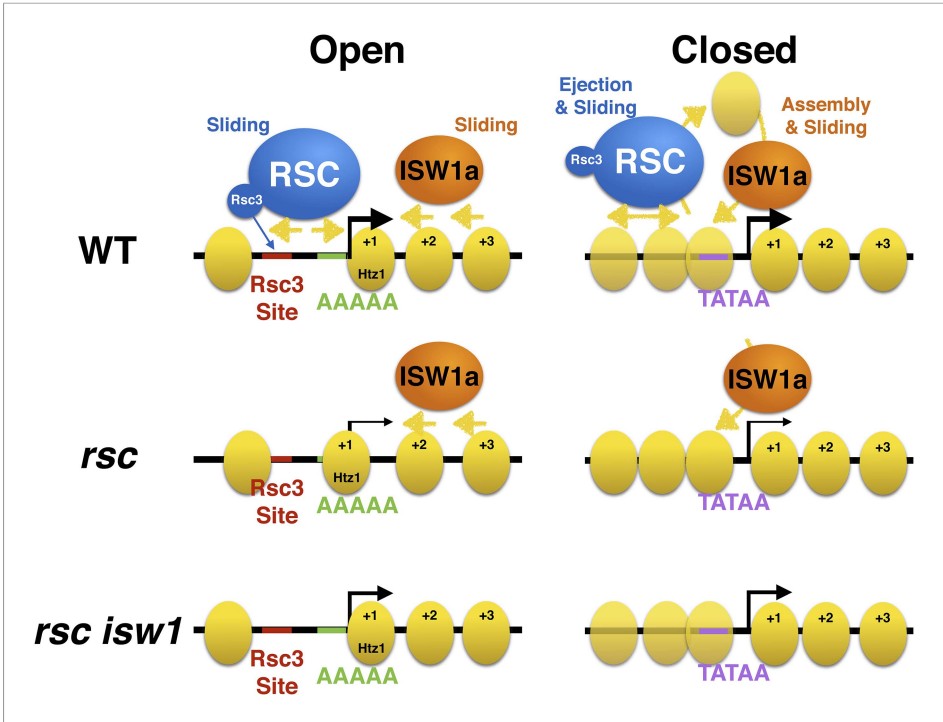

**Figure 7**. Model of action by RSC and ISWI remodelers at open and closed promoters. An open or structured promoter is depicted on the left with regularly spaced nucleosomes (yellow ovals) and a predominate NDR that frequently contains sequence elements (colored lines), including Rsc3 and Reb1 binding sites as well AT-rich sequence tracts unfavorable to nucleosome formation. Remodelers such as RSC (blue oval) help to maintain nucleosome deficiency, while ISWIa (orange oval) antagonizes by 'filling-in' the NDR. (Note: Rsc3 is not required for RSC activity nor is Rsc3 required for all RSC recruitment.) In the absence of RSC, this filling-in occurs and is conducted by ISW1a, as filling-in is not observed in *rsc isw1* double mutants. A closed or unstructured promoter is depicted on the right, evidenced by the lack of a clearly defined NDR and obscured promoter sequence elements, such as the TATA. Nucleosome density (or likelihood of occupancy) is depicted by the opacity of the nucleosomes. These promoters have increased nucleosome movement and histone turnover (yellow arrows), likely aided by chromatin remodelers such as RSC and ISWI, which eject or reposition nucleosomes, respectively. In the absence of RSC, nucleosome ejection is reduced, leading to higher nucleosome density (opaque nucleosomes) and a reduction in transcription. Additional loss of ISW1a may reduce the assembly/organization of nucleosomes in the promoter, partially restoring transcription.

are other remodelers that also act at promoters, we consider the interactions described herein as the strongest evidence to date exemplifying chromatin remodeler antagonism.

Evidence for RSC-Isw1 antagonism was revealed through two entirely independent unbiased genetic screens for suppression of RSC mutants. The first screen utilized an SGA method to identify suppressors of *rsc7Δ* and revealed *isw1Δ* as the strongest of four identified gene suppressors and the only gene with a chromatin-related function. Indeed, combinations of *rsc* mutants with mutations in chromatin factors are almost invariably lethal (*rsc4*-HDAC combinations are a rare exception [*Kasten et al., 2004*]). The second screen—involving *rsc2* suppression by histone mutations—yielded a small set of mild suppressors in histone H3 and one suppressor of moderate strength, H4 RH17,18CY. This region of the H4 tail is known as the 'basic patch'—an epitope of known importance for the binding and activity of several chromatin-modifying factors including ISWI, Sir3, and Dot1 (*Clapier et al., 2001*; *Clapier et al., 2002*; *Altaf et al., 2007*; *Fingerman et al., 2007*). Further genetic work focused the impact of this mutation on ISWI function, then on Isw1 function, and finally on Isw1a function (as opposed to the compositionally distinct Isw1b complex). Notably, combinations of *isw1Δ* with Swi/Snf mutations did not confer suppression, indicating specificity for suppressing RSC function. Taken together, two independent genetic screens, combined with multiple additional genetic approaches, identify a specific suppression relationship between the RSC complex and the Isw1a complex.

This RSC-ISW1a suppression is also consistent with a recent report that loss of Isw1a complex can suppress the phenotypes of *gcn5Δ* mutations combined with loss of another H3 acetyltransferase, Sas3 (*Lafon et al., 2012*). It is possible that the *isw1Δ* and *ioc3Δ* suppression of *gcn5Δ sas3Δ* may be partially due to reducing the phenotypic effects of reduced acetylation by reducing RSC activity, since RSC function is partially dependent on acetylation (*VanDemark et al., 2007*).

We then explored whether this suppression relationship resulted from opposing roles of the two remodelers for regulating chromatin structure. A role for RSC at maintaining proper chromatin structure was previously demonstrated through the use of the strong *Sth1* degron allele (*Parnell et al., 2008*; *Hartley and Madhani, 2009*) and other RSC alleles (*Badis et al., 2008*; *Ganguli et al., 2014*). Loss of RSC function results in a gain of nucleosome density across Pol III genes and at the NDR of many Pol II genes. A clear observation here is the 'leftward' shift of the +1 and subsequent nucleosomes towards the NDR. This is consistent with (and extends) published models that RSC maintains the NDR by moving and/or ejecting nucleosomes from the TSS. Our work suggests that this movement and 'fill in' is, at least in part, performed by the ISWI family of remodelers, as we have demonstrated a reduction of the 'fill in' in the *rsc isw1* double mutant. Cells lacking *ISW1* alone exhibit modest changes in the coding region (*Ni et al., 2010*; *Tirosh et al., 2010*; *Yen et al., 2012*), which may, in part, be due to the loss of the ISW1b complex, which is thought to act primarily in the coding region, as opposed to the ISW1a complex that acts at promoters (*Morillon et al., 2003*). Our work here provides the first molecular examination of *rsc isw1* double mutants (prompted by our genetic suppression relationships) demonstrating antagonism between these remodelers regarding the depth of the NDR, the occupancy and positioning of the +1 nucleosome, and the phasing of proximal nucleosomes in the coding region (*Figure 7*).

The clustering of gene promoters into different classes based on their chromatin response to the loss of RSC function also revealed an interesting insight regarding the organization of promoter chromatin (*Figure 7*). More responsive genes have an open/structured promoter, with a classic −1, NDR, and +1 nucleosome at uniform positions with respect to the TSS. These patterns are evolutionarily conserved and partially imposed by sequence (*Ioshikhes et al., 2006*; *Tsankov et al., 2011*), where the open promoters demonstrate a higher enrichment of nucleosome exclusion sequences, such as tracts of AAAAA, and illustrated by the strong NDR in the prediction model. While sequence alone cannot entirely dictate chromatin structure (*Zhang et al., 2009*), chromatin remodelers like RSC are able to reinforce the NDR by moving nucleosomes out of the NDR. The increased likelihood of Rsc3 or Reb1 binding sites occurring within the NDR may help recruit RSC or other factors to promoters that require nucleosome sliding or ejection activity to maintain this open architecture (*Badis et al., 2008*; *Hartley and Madhani, 2009*). However, we note that the transcriptional output from these open promoters appears less affected following RSC loss than at closed/structured promoters (see below).

In contrast, the genes that lack a uniform chromatin response to RSC loss tend to have a closed or covered promoter, where nucleosomes are not uniformly positioned with respect to the TSS (*Figure 7*). This is not to say that these promoters have no chromatin structure at all; rather, each promoter has a unique chromatin structure that is not uniformly identical in phasing. In composite measurements, such as those presented in *Figure 5B*, these promoters appear to have little chromatin structure, when, in reality, they simply lack consensus structure. These promoters have an increased likelihood to have a TATA box and other transcription factor binding sites, whose access may be regulated by the partial occlusion by nucleosomes (*Ioshikhes et al., 2006*; *Tirosh and Barkai, 2008*). Here, Isw1a may function to help assemble/mature and properly space nucleosomes at these promoters to repress transcription, which then increases their reliance upon remodelers such as RSC and/or SWI/SNF to expose these binding sites for proper activation. Hence, these promoters would have an increased presence of both activating and repressing chromatin remodelers, as well as histone turnover, both of which we observe (*Figure 7*). This continual state of flux, as well as lack of uniformity, may help explain why we observe little collective change in the chromatin structure in the absence of RSC function, while also observing a greater reliance on RSC function to maintain an active transcriptional status.

Taken together, our study provides the first evidence for an antagonistic relationship between RSC and ISWI, showing the genetic suppression of growth phenotypes and the lessening of chromatin impact due to the loss of RSC function. These effects are revealed on a genome-wide scale and further reveals that particular promoter chromatin architectures can influence the degree of impact.

These results reveal the different strategies chromatin used by genes for maintaining and regulating genic transcription through the use of promoter architecture, DNA accessibility, and the antagonism between complexes that act on promoter chromatin.

# Materials and methods

## Media, genetic methods, and strains

Rich media (YPD), synthetic complete (SC), minimal synthetic defined (SD), and sporulation media were prepared by standard methods. Standard procedures were used for transformations, sporulation, and tetrad analysis. All strains are derivatives of S288C, and full strain genotypes are listed in *Supplementary file 1*. Plasmids used are listed in *Supplementary file 2*. Null mutations in *ISW1* or *IOC3* were obtained from Invitrogen (Carlsbad, CA) and crossed in, or made by PCR disruption, and confirmed by PCR and complementation.

## Genetic screens for suppressors of *rsc2* and *rsc7* temperature-sensitive mutants

To isolate mutations in Histone H3 or Histone H4 that could suppress an *rsc2* TS⁻ mutant, p1411 [*HHT2-HHF2.TRP1*] was mutagenized with hydroxylamine and transformed into YBC2140 (*rsc1Δ rsc2-V457M hht1Δ-hhf1Δ hht2Δhhf2Δ* [*HHT2-HHF2.URA3*]). Approximately 20,000 transformants were plated to SC-TRP + 5FOA medium, incubated at 33°C, and screened for colony growth. Resident plasmids conferring suppression were isolated, retransformed, and sequenced.

The SGA screen was performed by mating *rsc7Δ* [*RSC7.URA3*] (YBC2039) with the yeast haploid deletion set (BY4741) from Invitrogen and isolating double mutants as described in *Wilson et al. (2006)*. Double mutants were scored for the ability to grow at 35°C following *RSC7* plasmid loss on 5FOA.

## RSC and ISW1 ChIP analysis

*RSC8*, *SNF2*, and *IOC3* genes were tagged endogenously with 13xMyc tags as described (*Longtine et al., 1998*). Yeast strains were grown in either rich media (YPD) or minimal media (SD) and ChIP performed from both samples as described previously (*Parnell et al., 2008*). ChIP eluates and input DNA were labeled with either Cy5 or Cy3, and two biological replicates of each were hybridized to Agilent 244K microarrays. The ChIP efficiency was better in cells grown in SD media, perhaps due to increased cross-linking efficiency (rich media may inherently have a quenching effect relative to minimal media). Comparison between YPD- and SD-derived occupancies revealed little differences besides the relative scale of enrichment; therefore, all analysis was performed using the SD data.

For ChIP sequencing, the Rsc8-Myc, Ioc3-Myc, and Snf2-Myc strains were used in addition to a strain expressing Sth1 tagged with 2xFlag under a Met25 promoter. ChIP conditions were similar to those used previously, except chromatin was liberated by micrococcal nuclease. Immunoprecipitated products and corresponding input were assembled into a library using Illumina protocols. Library products were size-selected for mono-nucleosomes prior to paired-end sequencing (36 bp for Rsc8, 50 bp for remainder) using Illumina sequencers.

## HybMap RNA preparation

The HybMap microarray was custom designed to represent genes with a range of RSC and ISW1 occupancies and included 448 coding genes, 93 tRNA genes, and 52 non-coding genes. Gene regions were extended by either 300 bp (coding and non-coding) or 150 bp (tRNA). Probes were selected from a pool of tiled 60 mers and adjusted for length to match melting temperatures as necessary. Both strands for each probe were included in the design. Probes have a mean spacing of ~50 bp. As a control, 502 probes with sequences from zebrafish were included as non-hybridizing control probes; sequences were confirmed not to have significant homology to yeast sequences. Microarray designs were submitted to Agilent Technologies for production as 4 × 44K arrays. Total RNA from three biological replicates was prepared from each yeast strain, hybridized to the array, and detected as described (*Dutrow et al., 2008*).

## Nucleosome preparation

Yeast strains were grown under degron-inducing conditions, and mono-nucleosomal DNA was isolated as described previously (*Parnell et al., 2008*). DNA fragments were size-selected by agarose

gel electrophoresis, purified, and labeled with either Cy3 or Cy5. Labeled DNA from three biological replicates was co-hybridized to Agilent 244K microarrays for each strain. For sequencing, mono-nucleosomal DNA was prepared into a library using Illumina kits and subjected to paired-end 50 bp sequencing.

## Bioinformatic analysis

Raw microarray data were quantile normalized, averaged, median scaled, and assigned to genomic coordinates. For the HybMap protocol, probe values were median scaled to the median intensity from the zebrafish control probes. Probe sequences were mapped to the *Saccharomyces cerevisiae* genome version 64 (Saccharomyces Genome Database). Gene transcript models were based on whole-genome transcriptome data (*Xu et al., 2009*). Transcription start and stop sites were generated from processed transcriptome data and compared and merged with published transcript models. Transcripts with discrepancies were manually curated using published occupancy maps for nucleosome and promoter initiation factors as guides. This resulted in a list of 5338 high-quality transcript models. For ChIP sequencing data, including published data sets obtained through NCBI, raw Fastq alignments were aligned using Novoalign and processed using the MACS2 software (https://github.com/taoliu/MACS) to generate fold enrichment data.

Most analysis was performed using BioToolBox (https://github.com/tjparnell/biotoolbox). Cluster analysis was visualized with Java Treeview (*Saldanha, 2004*). Statistics and graphs were generated with GraphPad Prism (GraphPad Software, Inc.). Intersection analysis was performed with the USeq package (*Nix et al., 2008*). ChIP enrichment FDR values were calculated using MACS2.

## Supplemental information

Supplemental figures and files are available. Raw microarray and sequencing data are available at GEO under accession number GSE65594.

## Acknowledgements

We thank Sharon Dent, Toshio Tsukiyama, and Kevin Struhl for strains and/or plasmids. We thank HHMI and NIH (GM60415) for funding. We thank Melinda Angus-Hill for insight into *isw1Δ* suppression of *rsc3-3*; Shan-Fu Wu for assistance in strain generation; and members of the Cairns lab, Mahesh Chandrasekharan, and Tim Formosa for discussion.

## Additional information

### Funding

| Funder | Grant reference | Author |
| --- | --- | --- |
| Howard Hughes Medical Institute (HHMI) | GM60415 | Bradley R Cairns |
| National Institutes of Health (NIH) | GM60415 | Bradley R Cairns |

The funders had no role in study design, data collection and interpretation, or the decision to submit the work for publication.

### Author contributions

TJP, AS, Conception and design, Acquisition of data, Analysis and interpretation of data, Drafting or revising the article, Contributed unpublished essential data or reagents; BGW, Acquisition of data, Analysis and interpretation of data, Contributed unpublished essential data or reagents; BRC, Conception and design, Analysis and interpretation of data, Drafting or revising the article

## Additional files

### Supplementary files

• Supplementary file 1. Table of yeast strains. List of yeast strains and their genotypes used in this study.

• Supplementary file 2. Table of plasmids. List of plasmid names and their sources used in this study.

### Major dataset

The following dataset was generated:

| Author(s) | Year | Dataset title | Dataset ID and/or URL | Database, license, and accessibility information |
|---|---|---|---|---|
| Parnell TJ | 2015 | The Chromatin Remodelers RSC and ISW1 Display Functional and Chromatin-based Promoter Antagonism | http://www.ncbi.nlm.nih.gov/geo/query/acc.cgi?acc=GSE65594 | Publicly available at NCBI Gene Expression Omnibus (GSE65594). |

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
