## [Decision Letter]

Thank you for sending your work entitled “RSC and ISW1 Chromatin Remodelers Display Functional and Chromatin-based Promoter Antagonism” for consideration at *eLife*. Your article has been favorably evaluated by Aviv Regev (Senior editor) and three reviewers, one of whom is a member of our Board of Reviewing Editors.

The Reviewing editor and the other reviewers discussed their comments before we reached this decision, and the Reviewing editor has assembled the following comments to help you prepare a revised submission.

This paper addresses the functional antagonism between the RSC and ISW1a remodeling complexes, a subject that is of emerging interest in the chromatin field. By using a synthetic genetic array (SGA) approach to screen for suppressors of *rsc* mutants, the authors discovered that another remodeling protein, Isw1, specifically ameliorates the *rsc* mutant phenotypes. Nucleosome mapping studies revealed global effects of RSC and ISW1a complexes acting against each other in shaping the promoter chromatin architecture. This work appears to be a promising candidate for *eLife*. There are, however, a number of specific questions and concerns that need to be addressed.

1) It was somewhat surprising that the general changes in nucleosome arrangement were accompanied by very few changes in RNA levels (Figure 4; Figure 4—figure supplement 2). For the few genes (e.g., MTH1, GOS1, SNF6, HIS4) that do show clear gene expression changes upon suppression by *ioc3Δ* or *isw1Δ,* the authors should also reveal the corresponding nucleosome maps so that readers can assess the relationship between nucleosome occupancy/position and transcription, and whether there is any correlation between the chromatin and transcription in this respect. Whatever the correlation, the issue should be articulated in the text, and possible explanations provided. Likewise it would be useful to know where the functionally affected genes fall in the categories of open and closed promoters. Do any rules emerge, or is it just random?

2) There appears to be a contradiction between Figures 4 and 6. In Figure 4, it appears that clusters 5 and 6 have the lowest RSC occupancy, whereas in Figure 6, it appears that clusters 5 and 6 have the highest RSC occupancy. This point needs to be clarified.

3) The summary diagram Figure 7 “Open” can be easily interpreted, but Figure 7 “Closed” is confusing.

---

## [Author Response]

*1) It was somewhat surprising that the general changes in nucleosome arrangement were accompanied by very few changes in RNA levels (*Figure 4*;*
Figure 4—figure supplement 2*). For the few genes (e.g., MTH1, GOS1, SNF6, HIS4) that do show clear gene expression changes upon suppression by* ioc3Δ *or* isw1Δ, *the authors should also reveal the corresponding nucleosome maps so that readers can assess the relationship between nucleosome occupancy/position and transcription, and whether there is any correlation between the chromatin and transcription in this respect. Whatever the correlation, the issue should be articulated in the text, and possible explanations provided. Likewise it would be useful to know where the functionally affected genes fall in the categories of open and closed promoters. Do any rules emerge, or is it just random*?

As requested, we examined the genic nucleosome profile analysis for the 20 genes identified as suppressed by either *isw1* or *ioc3*. Ideally, one could link changes in nucleosome positioning to expression by using high-resolution data to show that a nucleosome has moved with respect to an important cis-controlling element, and then infer that the movement might underlie the expression change. However, our profiles are derived from a microarray platform, and although these arrays proved to be very reproducible and provide great ‘class average’ maps of gene/promoter cohorts, their resolution does not allow us to state with confidence that a single nucleosome underlies the expression change. This is indeed a limitation worth discussing, so we have added text in the Results section to reflect this limitation, and we have also included how these genes partition into the open and closed categories. This is one reason why we did not focus our paper on these single genes, but instead focused the work on our clear genetic and genomic (cohort average) evidence for antagonism between these two Remodeler families and how this interplay affects promoter/gene architecture at cohorts of genes. Future work by us and others will indeed address their interplay at particular genes.

*2) There appears to be a contradiction between*
Figures 4 and 6*. In*
Figure 4*, it appears that clusters 5 and 6 have the lowest RSC occupancy, whereas in*
Figure 6*, it appears that clusters 5 and 6 have the highest RSC occupancy. This point needs to be clarified*.

In retrospect, we see how this was a major source of confusion, but it is easily explained and corrected. The six clusters in Figure 4 and Figure 6 actually involve different datasets, clustered separately, that just happened to both have six clusters. We thank the reviewers for noticing this, as we now see how a reader could have easily thought that these were the same datasets. In actuality, these two figure panels represent independent k-means cluster organizations of different data. Figure 4 represents the k-means cluster organization of the RSC and ISW1a occupancy based on the microarray data, and was provided entirely for illustrative purposes to simply show the similarity of RSC and ISW1a occupancy. Figure 6 displays instead the k-means clustering of the rsc null/RSC nucleosome ratio generated (and shown) in Figure 5, and thus bear no relationship to those derived in Figure 4. To reduce confusion, the cluster labels have been removed from Figure 4; these labels and groups are never mentioned in the manuscript. Furthermore, we make this distinction clear in the text. Additionally, some references to cluster groups in the text now explicitly reference to those clusters derived in Figure 5, to help guide the reader.

*3) The summary diagram*
Figure 7
*“Open” can be easily interpreted, but*
Figure 7
*“Closed” is confusing*.

We agree, and have revised the right side of the diagram figure fairly extensively. We have removed some of the action arrows with the same color as the remodeler, as we believe this was confusing as to what function these arrows portray. Instead, all action arrows are the same color as the nucleosomes to indicate what direction nucleosomes are moving.